

# CASPredict: a web service for identifying Cas proteins

Shanshan Yang[1], Jian Huang[2] and Bifang He[1,2]

[1] Medical College, Guizhou University, Guiyang, Guizhou Province, China
[2] Center for Informational Biology, University of Electronic Science and Technology of China, Chengdu, Sichuan Province, China

## ABSTRACT

Clustered regularly interspaced short palindromic repeats (CRISPR) and their associated (Cas) proteins constitute the CRISPR-Cas systems, which play a key role in prokaryote adaptive immune system against invasive foreign elements. In recent years, the CRISPR-Cas systems have also been designed to facilitate target gene editing in eukaryotic genomes. As one of the important components of the CRISPR-Cas system, Cas protein plays an irreplaceable role. The effector module composed of Cas proteins is used to distinguish the type of CRISPR-Cas systems. Effective prediction and identification of Cas proteins can help biologists further infer the type of CRISPR-Cas systems. Moreover, the class 2 CRISPR-Cas systems are gradually applied in the field of genome editing. The discovery of Cas protein will help provide more candidates for genome editing. In this paper, we described a web service named CASPredict (http://i.uestc.edu.cn/caspredict/cgi-bin/CASPredict.pl) for identifying Cas proteins. CASPredict first predicts Cas proteins based on support vector machine (SVM) by using the optimal dipeptide composition and then annotates the function of Cas proteins based on the hmmscan search algorithm. The ten-fold cross-validation results showed that the 84.84% of Cas proteins were correctly classified. CASPredict will be a useful tool for the identification of Cas proteins, or at least can play a complementary role to the existing methods in this area.

Corresponding author
Bifang He, bfhe@gzu.edu.cn

## INTRODUCTION

The CRISPR-Cas systems consist of clustered regularly interspaced short palindromic repeats (CRISPR) and CRISPR-associated (Cas) proteins (*Hille et al., 2018*). The first batch of CRISPR with dyad symmetry structure was detected by *Ishino et al. (1987)* in *Escherichia coli*. At that time, the lack of adequate DNA sequence data, especially for mobile genetic elements, made it nearly impossible to predict the biological function of these abnormal repeated sequences (*Ishino, Krupovic & Forterre, 2018*). Later, with the continuous development of genomics, *Barrangou et al. (2007)* experimentally demonstrated that the CRISPR and related Cas genes work together against phages. This experiment finally confirmed the function of the CRISPR-Cas system as a prokaryotic acquired immune system. Since then, the CRISPR-Cas systems have gradually become a research hotspot in the field of gene editing. The process of CRISPR-Cas acquired immune

**How to cite this article** Yang S, Huang J, He B. 2021. CASPredict: a web service for identifying Cas proteins. **PeerJ** 9:e11887
system is divided into three stages: (i) adaptation, where invading DNA is recognized by Cas proteins, fragmented and incorporated into the CRISPR array, and stored in the genome; (ii) expression, where the CRISPR array is transcribed and cut into mature CRISPR RNAs (crRNAs). (iii) interference, where foreign DNA is captured by taking advantage of the homology of the spacer sequence present in crRNA, and cleaved by a complex with Cas protein having nuclease activity (*Alkhnbashi et al., 2020*). It closely resembles the process of RNA interference in eukaryotes. Due to the precise targeting function, the CRISPR-Cas system has been gradually developed as an efficient gene editing tool. Moreover, different types of Cas proteins have different functions in the three stages of the CRISPR-Cas system. According to the function performed by the Cas proteins, the Cas proteins can be divided into core and auxiliary Cas proteins. The core Cas proteins mainly participate in the immune process of the CRISPR-Cas system, while the auxiliary Cas proteins play a role in other stages of the CRISPR-Cas immunity and assist in completing the immune process of the CRISPR-Cas system. At present, a total of 13 Cas core protein families have been identified and named, which are Cas1, Cas2, Cas3, Cas4, Cas5, Cas6, Cas7, Cas8 (large subunit), Cas9, Cas10 (large subunit), Cas11 (small subunit), Cas12, and Cas13 (*Koonin & Makarova, 2019*). These core Cas proteins play an important role in prokaryotic species to resist foreign nucleic acid molecules. However, the function and mechanism of some Cas proteins are still unclear, such as Cas4 protein, whose function is only predicted currently. Therefore, the function and mechanism of Cas proteins need to be further studied.

Based on the detailed sequence analyses and gene organization of the Cas proteins, CRISPR-Cas systems are divided into classes 1 and 2 according to the number of Cas proteins (*Ishino, Krupovic & Forterre, 2018*). Class 1 includes types I, III, and IV, which utilize multiple Cas protein units to form an effector complex together with the crRNA for recognizing and cleaving target strands. Class 2 includes types II, V, and VI, which utilize a single Cas protein as the effector (*Murugan et al., 2017*), and the common effectors are Cas9 (Csn1, II), Cas12a (Cpf1, VA), Cas12b (C2c1, VB), Cas13a (C2c2, VI-A), Cas13b (C2c6, VI-B) and Cas13c (C2c7, VI-C) (*Shmakov et al., 2017*). Due to the simple architecture of effector complexes, Class 2 CRISPR-Cas systems have more advantages in developing innovative gene editing technologies compared to Class 1. Effectively identifying Cas proteins can help biologists further infer the type of CRISPR-Cas systems (*Abby et al., 2014*). In addition, the discovery of Cas protein will provide more candidates for genome editing.

However, there are only 293 manually reviewed Cas proteins included in the Uniprot by our last count. Many new Cas proteins need to be discovered urgently. Therefore, it is necessary to find effective and computational methods to quickly identify Cas proteins from bacteria and archaea proteomic data. Some bioinformatics tools for the CRISPR-Cas systems have been released in the past few years, such as CRISPRFinder (*Grissa, Vergnaud & Pourcel, 2007*) and its recent descendant CIRSPRCasFinder (*Couvin et al., 2018*). The former is a web tool for identifying CRISPR arrays, while the latter supports identifying both CRISPR arrays and Cas genes. Besides, tools for identifying CRISPR arrays also include PILER-CR (*Edgar, 2007*), CRISPRDetect (*Biswas et al., 2016*), etc.

However, there are relatively few computational methods special for identifying Cas proteins. MacSyFinder adopted hidden Markov model (HMM) to predict the categories and component information of Cas proteins system (*Abby et al., 2014*). *Chai et al. (2019)* used the hmmscan similarity search algorithm to develop a web service, called HMMCAS, for identifying and annotating Cas proteins. Currently, support vector machine (SVM) as a machine learning method has been widely used in pattern recognition and classification in bioinformatics (*Ge, Zhao & Zhao, 2020*). Therefore, we built a web service named CASPredict, which is composed of two modules. The first module is an SVM-based predictor for identifying Cas proteins, and the second is based on the hmmscan search algorithm for annotating the function of Cas proteins. The CASPredict can facilitate the discovery of potential Cas proteins from unknown protein sequences. We expect that CASPredict can help researchers find more Cas protein candidates and provide more data resources for the research of CRISPR-Cas systems.

## MATERIALS & METHODS

### Datasets

In the training dataset, we used Cas proteins as the positive samples and non-Cas proteins as the negative samples. In total, we extracted 293 manually reviewed Cas proteins from Uniprot (*UniProt, 2015*) (see 293Cas-proteins.xlsx in the Supplemental Data) and checked each protein manually. Among the 293 Cas proteins, those with non-standard letters ("X", "B" or "Z") were first excluded. Besides, if the benchmark dataset contains a large number of redundant and highly similar samples, it may produce misleading and overestimated results (*Feng, Chen & Lin, 2016*). In general, a lower threshold of sequence identity might help reduce the bias due to sequence homology, and in principle, a more reliable training model could be obtained, but the size of the dataset should also be considered (*Wang et al., 2019*). Therefore, we removed redundant sequences by CD-HIT (*Fu et al., 2012*) program with a sequence identity cut-off of 30%. As a result, we obtained 155 Cas protein sequences as positive samples. The negative samples were constructed through the following steps: (i) randomly extract non-Cas proteins from Uniprot that have been manually reviewed; (ii) exclude sequences harboring ambiguous residues, such as "X", "B" or "Z"; (iii) select the non-Cas proteins with sequence similarity less than 40% (*Wang et al., 2020*) compared to positive samples by CD-Hit-2D (*Fu et al., 2012*) program for avoiding the inaccurate prediction results caused by the homology bias and redundancy between negative and positive samples; (iv) remain the length distribution of the protein sequence in the positive and negative samples the same. (v) keep the number of negative and positive samples the same for avoiding the bias of the classification model caused by the imbalance of the number of positive and negative samples (*Cui, Fang & Han, 2012*; *Daberdaku & Ferrari, 2018*). Finally, the training dataset consisted of 155 Cas proteins and 155 non-Cas proteins.

To objectively evaluate the predictive ability of the model, we extracted 64 "Cas proteins" from Uniprot that had not been manually reviewed. Simultaneously, we randomly selected 64 non-Cas proteins from Uniprot that had been manually reviewed but were not in the training dataset. In total, the independent testing dataset was composed of

64 "positive" samples and 64 negative samples. The training dataset and independent testing dataset can be downloaded from http://i.uestc.edu.cn/caspredict/download.html.

## Feature extraction

Transforming traditional biological problems into computable mathematical models is the premise of bioinformatics analysis, and this process will directly affect the predictive performance of the predictor (*He, Chen & Huang, 2019*; *Padilha et al., 2020*). Methods based on dipeptide composition (DPC) have been successfully applied to feature extraction of protein sequences (*Ding et al., 2016*; *Tang et al., 2016*). Therefore, we utilized 400 dipeptide components to encode each sequence in the training dataset. DPC reflects protein sequence characteristics by calculating the frequency of each dipeptide, which is defined as follows:

$$DPC(i) = \frac{y(i)}{\sum_{i=1}^{400} y(i)} \tag{1}$$

where $DPC(i)$ is the frequency of the *ith* ($i = 1, 2, …, 400$) dipeptide, and $y(i)$ represents the number of the *ith* dipeptide in a protein sequence.

## Feature selection

Firstly, the fselect.py program in the LIBSVM 3.24 toolkit (*Chang & Lin, 2011*) was employed to determine each feature's contribution to the predictive system. Each dipeptide feature thus corresponds to an F-score. Features with higher F-scores have the higher capacity to separate the two groups. It should be noted that the calculation of F-scores was based on the data for training in each of the ten-fold or leave-one-out cross-validation but not the training dataset (155 Cas proteins and 155 non-Cas proteins). The following operations were performed to acquire the optimal feature subset: (i) use F-score and SVM to select the optimal feature subset by comparing the five-fold cross-validation results of top 10, 20, 30, 40, …, 390, and 400 features (Features are ranked by F-score in descending order. Parameters are optimized by using the grid.py program (*Chang & Lin, 2011*)); (ii) according to the five-fold cross-validation results, select the feature subset with the highest accuracy (optimal feature subset) to build the SVM model; (iii) evaluate the SVM model with the data for testing in each of the ten-fold or leave-one-out cross-validation. Besides, it is possible to obtain different prediction performance based on different feature selection strategies, so we also used random forest and SVM to select the optimal features. The process of feature selection based on random forest and SVM was similar to the above process, except that the features were ranked by the mean decrease impurity calculated by random forest.

## Support vector machine

SVM is a powerful classification method that has been widely used for protein prediction (*Li et al., 2017*; *Lin et al., 2015*). The basic principle of SVM is to map the input vector to the high-dimensional space through the kernel function, construct the separation hyperplane with the largest spacing, and realize the classification of observations. Empirical studies have shown that the prediction performance of the radial basis kernel

function (RBF) is better than the linear function, polynomial function, and sigmoid function (*He et al., 2016*; *Yang et al., 2016*). In this paper, we downloaded the integration toolkit LIBSVM 3.24 (*Chang & Lin, 2011*) to implement the construction of the classification model based on SVM. In addition, we obtained the optimal error factor c and kernel function variance gamma by using the python script grid.py in LIBSVM 3.24.

## Hmmscan search

To further verify the reliability of the SVM model and annotate the function of the Cas protein, we collected multiple sequence alignments or seed alignments for Cas protein families from the TIGRFAMs (version 15.0) and Pfam (version 34.0) databases (*Haft, Selengut & White, 2003*; *Mistry et al., 2021*). Totally, the collection contained 157 Cas protein families, among which 101 were from TIGRFAMs, and 56 were from Pfam. The hmmbuild program was then used to construct profile HMMs from multiple sequence alignments with default parameters (*Finn, Clements & Eddy, 2011*). In the end, we used hmmpress to format the Cas protein profile HMM database into a binary format for hmmscan. The Cas protein profile HMMs can be downloaded from http://i.uestc.edu.cn/caspredict/download.html. The hmmscan (*Finn, Clements & Eddy, 2011*) was used to search each query sequence against the collection of Cas protein profiles and output ranked lists of the profiles with the most significant matches to the sequence.

## Prediction assessment

In the case of limited datasets, the predictive performance of the model is usually evaluated by N-fold cross-validation. In this work, we used ten-fold cross-validation and leave-one-out cross-validation to evaluate the SVM model performance, respectively. In ten-fold cross-validation, the 310 proteins (155 Cas proteins and 155 non-Cas proteins) in the training dataset were randomly divided into ten folds. One fold was used for testing and the remaining nine folds were used for training. Repeat the feature selection process (see in the feature selection section) until each fold was used for testing once. Every time this process was repeated, an accuracy will be obtained. Eventually, we got the average prediction accuracy of SVM models. The leave-one-out cross-validation can also be called 310-fold cross-validation, its implementation was similar to ten-fold cross-validation. The main difference was that the 310 proteins were randomly divided into 310 folds. Each fold was used once for testing, and correspondingly the remaining 309 folds were used for training. For each of 310-fold cross-validation, execute the feature selection process (see in the feature selection section). After repeating the process 310 times, we got the average performance of SVM models.

To assess the performance of the models, we used the sensitivity (Sn), specificity (Sp), accuracy (Acc) and Matthews correlation coefficient (MCC) as evaluation indicators, which are defined as follows:

$$Sn = \frac{TP}{TP + FN} \tag{2}$$

$$Sp = \frac{TN}{TN + FP} \tag{3}$$

$$Acc = \frac{TP + TN}{TP + FN + FP + TN} \tag{4}$$

$$MCC = \frac{TP \times TN - FP \times FN}{\sqrt{(TP + FP)(TP + FN)(TN + FP)(TN + FN)}} \tag{5}$$

where TP, FP, TN and FN mean the amount of true positives, false positives, true negatives, and false negatives, respectively. For a prediction model, Sn and Sp focus on estimating its ability to identify positive and negative samples, while Acc and MCC reflect its comprehensive ability to identify both positive and negative samples (*Wang et al., 2020*). In addition, we also introduced another evaluation indicator called the area under the receiver operating characteristic (ROC) curve (AUC), which represents the probability that a positive sample's decision value is greater than a negative sample's decision value. In short, the larger the AUC value of the prediction model, the higher the classification accuracy of the prediction model.

## Evaluation model construction and comparison with HMMCAS

We selected all the proteins belonging to Cas13a family and the Cas12b protein from 155 Cas proteins as the new positive testing dataset (there were seven Cas proteins, including six Cas13a proteins and one Cas12b protein.), and randomly selected seven non-Cas proteins from 155 non-Cas proteins as the negative testing dataset. The remaining Cas proteins (148 Cas proteins and 148 non-Cas proteins) from 155 Cas proteins and 155 non-Cas proteins were used for training the SVM-based evaluation model. The F-score of each feature was calculated based on the training dataset, and a grid search strategy was also applied on the training dataset to seek for the best feature number, the error factor c and kernel function variance gamma. The training dataset and testing dataset for the evaluation model can be obtained from Supplemental Data eva_dataset.xlsx. The testing dataset was used to evaluate the performance of the SVM-based evaluation model and HMMCAS.

## Final model construction

The final classification model was based on the training dataset (155 Cas proteins and 155 non-Cas proteins). In the final model building process, we put the feature into the initially empty set in descending order by F-score one by one, and calculated the accuracy of each feature subset by five-fold cross-validation when adding a feature. Then, we selected the feature subset with the highest accuracy as optimal feature subset. It should be noted that the F-score of each feature was calculated based on the training dataset, and a grid search strategy was also applied on the training dataset to seek for the best feature number, the error factor c and kernel function variance gamma. The model construction and evaluation were performed at a computational server (Sugon I840-G20, Dawning Information Industry Co., LTD., Beijing, China).
**Table 1 Performances of SVM-based models trained with different feature selection methods and features.**

| Feature selection method | Feature | Sn (%) | Sp (%) | Acc (%) | MCC |
|---|---|---|---|---|---|
| NA | DPC | 81.34 | 83.60 | 81.61 | 0.64 |
| F-score | ODPC | 83.71 | 86.77 | 84.84 | 0.70 |
| Random forest | ODPC | 83.67 | 86.47 | 84.19 | 0.70 |

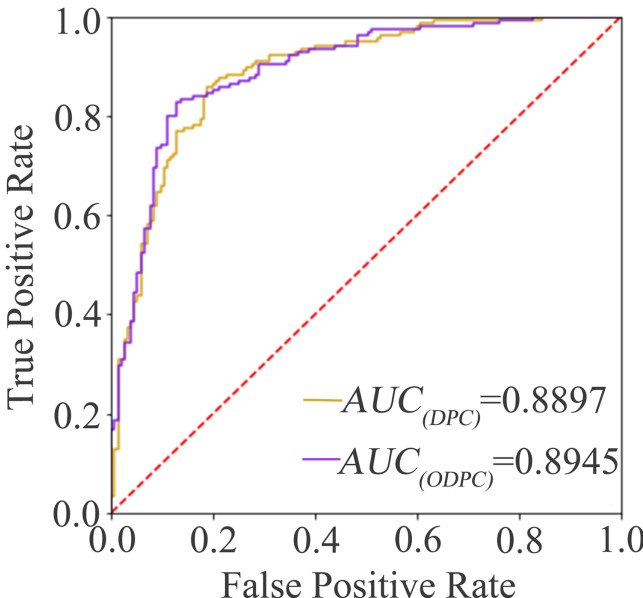

**Figure 1 The ROC curve graph of the prediction model based on DPC and ODPC.** The AUC value of DPC and ODPC is 0.8897 and 0.8945, respectively.

## RESULTS

### The construction of prediction model and performance

The average accuracy of the ten-fold cross-validation based on the F-score and random forest for feature selection was 84.84% and 84.19% (see Table 1), respectively, and the average accuracy of the leave-one-out cross-validation based on the F-score was 83.23%. The result shows that the models achieve a promising performance, and the method of using F-score for feature selection is superior to random forest. We called the optimal features obtained through feature selection as optimal DPC (ODPC). In the ten-fold cross-validation, the models were built based on the DPC and ODPC (ODPC selected by using F-score and SVM), respectively. The results in Table 1 show that the accuracy of models constructed with DPC and ODPC is 81.61% and 84.84%, respectively.

It demonstrates that the feature selection technique is helpful to improve the prediction performance. Besides, we described the prediction performance of the models intuitively by drawing the ROC curve. Figure 1 shows the ROC curves of the models built with DPC and ODPC, respectively. The AUC of the predictive model based on ODPC is approximately 0.89, which shows an impressive prediction performance.
**Table 2 The prediction performances of various machine learning methods.**

| Machine learning methods | Sn (%) | Sp (%) | Acc (%) | MCC |
|---|---|---|---|---|
| Support vector machine | 83.71 | 86.77 | 84.84 | 0.70 |
| Naïve Bayes | 81.29 | 78.06 | 79.68 | 0.59 |
| Logistic function | 65.16 | 70.97 | 68.06 | 0.36 |
| Decision tree J48 | 60.65 | 63.87 | 62.26 | 0.25 |

Therefore, we built the final SVM model by using the optimal feature subset with 167 dipeptide features (see optimal-features.docx in the Supplemental Data).

## Comparison with other machine learning methods

Comparing the proposed method with other existing methods is a necessary way to verify whether the method has advantages. To compare with SVM-based classifier, we constructed classifiers based on Naïve Bayes, Logistic Function, and Decision Tree J48 for ODPC, respectively. These classifiers were implemented in WEKA (*Frank et al., 2004*). As the ten-fold cross-validation results shown in Table 2, the average accuracy of the SVM-based classifier is about 5.16%, 16.78%, and 22.58% higher than that of Naïve Bayes, Logistic Function, and Decision Tree J48 classifier, respectively. It demonstrates that the SVM-based method has a better performance compared with other machine learning methods.

## Comparison with HMMCAS

To compare with HMMCAS, a new SVM-based evaluation model was built by using 296 proteins data (148 Cas proteins and 148 non-Cas proteins) with 161 dipeptide features. The evaluation model and HMMCAS were used to analyze the seven Cas proteins (six Cas13a proteins and one Cas12b protein.) and seven non-Cas proteins. A total of seven Cas proteins and six non-Cas proteins were correctly identified by the new SVM-based evaluation model when the threshold to distinguish between predicted positives and negatives was set to 0.5. However, only one Cas protein was correctly identified but all non-Cas proteins were correctly identified by HMMCAS (the parameter settings of HMMCAS are shown in Supplemental Data hmmcas-parameter.docx). It verifies that the SVM-based model is effective for Cas protein prediction.

## Web service

For the convenience of users, the SVM-based model built with ODPC was implemented into an online web service, called CASPredict, which is freely available at http://i.uestc.edu.cn/caspredict/cgi-bin/CASPredict.pl. In the online web service, the common gateway interface script was written using Perl. As shown in Fig. 2, users submit protein sequences in FASTA format. After clicking "Predict" button, the predictive results will be displayed in a table (see Fig. 3). In the prediction process, CASPredict first identifies the sequences submitted by users based on SVM, and then uses hmmscan homology search algorithm in
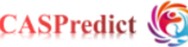

| Home | **Predict** | Citation | Download | Help |

On the predict page, you can submit some sequences for prediction. **Please Pay attention:** the CASPredict tool only supports protein sequence in FASTA format. Line breaks should be carried out between each sequence, and there should be no blank characters such as spaces or line breaks within each sequence. If you only have raw sequence information, you must construct a simple FASTA format for the raw sequence before making prediction. The format is as follows: "**>+name+\n+sequence**", where the "**name**" is the name you specify for the sequence and it can be made up of any non-null characters, as shown in the example. If you upload a sequence file, please name the file suffix as **.txt** or **.fasta** format. Besides, only the standard IUPAC one-letter codes for the amino acids (A,C,D,E,F,G,H,I,K,L,M,N,P,Q,R,S,T,V,W,Y) are supported. **Note:** if you want to submit genome sequences for Cas proteins prediction, please first use ORF Finder (https://www.ncbi.nlm.nih.gov/orffinder/) to transform the genome sequences into protein sequences and then use CASPredict.

**Furthermore, the threshold to distinguish between predicted positives and negatives (*tp*)** ranges from 0 to 1. However, it is set to 0.5 by default. That is to say, a sequence will be predicted to be a Cas protein if the probability is 0.5 or higher. You can adjust the threshold according to your own needs. Here we go!

**Enter a set of protein sequences in the text area below:**

```
>Q97TX9
MITEFLLKKKLEEHLSHVKEENTIYVTDLVRCPRRVRY
ESEYKELAISQVYAPSAILGDILHLGLESVLKGNFNAET
EVETLREINVGGKVYKIKGRADAIIRNDNGKSIVIEIKTS
RSDKGLPLIHHKMQLQIYLWLFSAEKGILVYITPDRIAE
YEINEPLDEATIVRLAEDTIMLQNSPRFNWECKYCIFSV
ICPAKLT
```

Or upload a sequence file: [ ] Browse...

Set the *tp*: [0.5]

[Example] [Reset] [Predict]

**Figure 2** Prediction page of CASPredict web service at **http://i.uestc.edu.cn/caspredict/cgi-bin/CASPredict.pl**.

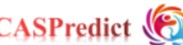

| Home | **Predict** | Citation | Download | Help |

All the results are stored in the table. You can click **Number, Length, or Probability** to sort the results in ascending or descending order. The "Probability" column shows the probability that the query sequence is predicted to be a Cas protein, which is obtained based on the machine learning method of SVM. When the "Yes/No" column is "Yes", it indicates that the sequence is predicted to be a Cas protein, and CASPredict will further use hmmscan to search this sequence against Cas protein family HMMs. If the "Description" column shows that the query protein is any one of Cas1, Cas2, Cas3, Cas4, Cas5, Cas6, Cas7, Cas8 (large subunit), Cas9, Cas10 (large subunit), Cas11 (small subunit), Cas12 and Cas13, indicating that the query protein is a core Cas protein. You can click the ❷ after each column name to get the meaning of each column name in the result table.

| Number ⇕ | Query ❷ | Length ⇕ | Probability ⇕ | Yes/No ❷ | Model ❷ | Description ❷ | E-value ❷ |
|---|---|---|---|---|---|---|---|
| 1 | sp\|Q97TX9\|CAS4_SACS2 | 202 | 0.94 | Yes | TIGR00372 | cas4: CRISPR-associated protein Cas4 | 4.5e-31 |
| 2 | sp\|Q57829\|CS3HD_METJA | 244 | 0.94 | Yes | TIGR01596 | cas3_HD: CRISPR-associated endonuclease Cas3-HD | 4.4e-24 |
| 3 | sp\|P55733\|SYRM2_SINFN | 339 | 0.15 | No | NA | NA | NA |

**Figure 3** The result page returned from CASPredict.

HMMER3.1 (*Finn et al., 2015*) to search the sequences of Cas proteins that recognized by SVM against the Cas protein profile HMMs (using the gathering threshold).

## Evaluation of CASPredict

To evaluate CASPredict, we constructed an independent testing dataset. The "positive" testing dataset contained 64 "Cas proteins" from Uniprot that had not been manually reviewed. The negative testing dataset contained 64 non-Cas proteins that were not used in the training dataset. CASPredict was tested with the independent testing dataset. Results from the "positive" testing dataset showed that 46 proteins were predicted to be possible Cas proteins with a positive rate of 71.88% when the threshold to distinguish between predicted positives and negatives (tp) was set to 0.5. The results from the negative testing dataset showed a positive rate of 9.38% (6 of the 64 proteins were predicted to be Cas proteins), which is significantly lower than the results from the "positive" testing dataset ($p < 0.05$, chi square test).

## DISCUSSION

The identification of Cas proteins is of great significance in promoting the classification of the CRISPR-Cas systems and discovering new candidate tools for genome editing. In this paper, we proposed a machine learning method based on SVM to identify Cas proteins from bacteria and archaea. The sequence representation method based on dipeptide composition was applied to feature encoding, and the feature selection technique based on F-score was applied to select the optimal feature subset. After feature selection, we obtained 167 features as ODPC, which were used to build the SVM model. The result of ten-fold cross-validation shows that the accuracy of our model is superior to other models based on Naïve Bayes, Logistic Function, and Decision Tree J48, respectively. It is concluded that the SVM-based method has the excellent predictive effect compared with other machine learning methods. Our model reached an accuracy of 84.84% with 0.70 MCC, 83.71% sensitivity, and 86.77% specificity, respectively.

Since we collected all the Cas proteins that had been manually reviewed from Uniprot, it is difficult to find a reliable dataset other than the positive samples from the training dataset (155 Cas proteins). Therefore, we utilized part of the existing dataset as the testing dataset to compare the performance of HMMCAS and the SVM-based evaluation model. Through checking the models in HMMCAS, we found that most of the proteins we collected had been utilized for the HMM model construction. Using these sequences would greatly overestimate the performance of HMMCAS. We further found that Cas13a family and Cas12b protein were not included in the training dataset of HMMCAS. Therefore, we selected all Cas13a family and Cas12b protein (totally seven sequences) and seven randomly picked negative sequences as a testing dataset to test the performance of HMMCAS. We constructed a new SVM evaluation model based on a new training dataset which did not include these 14 sequences and then used these 14 sequences to test the performance. The results showed that seven Cas proteins and six non-Cas proteins were

correctly identified by the new SVM evaluation model. However, only one Cas protein was correctly identified but all non-Cas proteins were correctly identified by HMMCAS. It verifies that the SVM-based model is effective for Cas protein prediction, but it is worth noting that the specificity of SVM-based model is inferior to HMMCAS.

Although Cas proteins belong to different families or subfamilies, they have similar sequence properties. Therefore, all Cas proteins should have some simple or highly complex hidden similarities. Based on such simple or complex similarities, we could make cross-class prediction of Cas proteins prospectively. HMM is frequently used for studying the hidden patterns in an observed sequence or sets of observed sequences. Although HMM has a broad range of usability in studying homology of protein sequences, it is a traditional probabilistic model in principle. Compared with nonlinear machine learning models, HMM is relatively weak in exploring the high-dimensional nonlinear characteristics of data. The results of this paper also found that HMM had little predictive effect on Cas13a and Cas12b protein sequences in the absence of Cas13a and Cas12b protein sequences. Under the same conditions, the SVM-based predictive model can accurately predict Cas13a and Cas 12b proteins based on the information of other types of Cas protein sequences. It shows that compared with HMM, SVM can better extract high-dimensional nonlinear characteristics of cross-class Cas protein sequences.

We also constructed an independent testing dataset with 64 "Cas proteins" and 64 non-Cas proteins to evaluate the generalization ability of CASPredict. The positive rate of 71.88% for the "positive" testing dataset was significantly higher than that for the negative testing dataset ($p < 0.05$, chi square test). Proteins in the "positive" testing dataset may have a higher possibility to be Cas proteins, since there is clear experimental evidence for the existence of the proteins though not manually reviewed. Each protein in the negative testing dataset should have a lower possibility to be a Cas protein, since it does not belong to Cas protein family. However, 64 "Cas proteins" are not really reliable. Therefore, the independent testing dataset is flawed.

In order to facilitate the use of the model for prediction, we implemented the SVM-based model into an online web service called CASPredict. It is easy to operate and freely available at http://i.uestc.edu.cn/caspredict/cgi-bin/CASPredict.pl. CASPredict further uses hmmscan to search the sequences against Cas protein family HMMs when the query sequence is identified to be a Cas protein by the SVM-based model. If the query sequence is recognized as a Cas protein using SVM, and a model that matches the sequence can be found in Cas protein profile HMMs, it proves that the predictive result based on SVM is highly reliable.

The CASPredict proposed in this paper is quite promising and holds a potential to become a useful tool for identifying Cas proteins, or at least can play a complementary role to the existing methods in this area. In the future, we will continue to collect new Cas proteins, further combine genomic and proteomic data to identify the putative Cas proteins, and provide an important basis for CRISPR-Cas system classification or finding candidate scissors for gene editing.

## ACKNOWLEDGEMENTS

The authors are grateful to the anonymous reviewers for their valuable suggestions and comments, which will lead to the improvement of this paper.

### Code Availability

The source code under a MIT license is freely available at https://github.com/shanshan1996/caspredict.

### Funding

This work was supported by the National Natural Science Foundation of China (grant number: 61901130, 61901129 and 62071099), the 2018 Talent Research Program of Guizhou University (Grant Numbers: (2018) 54 and (2018) 55), the Science and Technology Plan Project of Guizhou Province of China (Grant Numbers: (2018) 5781, [2020] 1Y407 and [2020] 1Y345), and the China Postdoctoral Science Foundation Grant (Grant Numbers: 2019M653369 and 2019M660236). The funders had no role in study design, data collection and analysis, decision to publish, or preparation of the manuscript.

### Grant Disclosures

The following grant information was disclosed by the authors:
National Natural Science Foundation of China: 61901130 and 62071099.
Guizhou University: (2018) 54.
Science and Technology Plan Project of Guizhou Province of China: [2020] 1Y407.

### Competing Interests

The authors declare that they have no competing interests.

### Author Contributions

- Shanshan Yang conceived and designed the experiments, performed the experiments, analyzed the data, prepared figures and/or tables, authored or reviewed drafts of the paper, and approved the final draft.
- Jian Huang conceived and designed the experiments, authored or reviewed drafts of the paper, and approved the final draft.
- Bifang He conceived and designed the experiments, analyzed the data, authored or reviewed drafts of the paper, and approved the final draft.

### Data Availability

The source code is available at GitHub: https://github.com/shanshan1996/caspredict.
The data is available in the Supplemental File.

### Supplemental Information

Supplemental information for this article can be found online at http://dx.doi.org/10.7717/peerj.11887#supplemental-information.

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
