# Peer review of "CASPredict: a web service for identifying Cas proteins"

_PeerJ, doi:10.7717/peerj.11887_

## Round 0.1 · original submission · Major Revisions

Dear Drs. Yang and colleagues:

Thanks for submitting your manuscript to PeerJ. I have now received two independent reviews of your work, and as you will see, the reviewers raised some concerns about the research. Despite this, these reviewers are optimistic about your work and the potential impact it will lend to research on informatics approaches to identifying Cas proteins. Thus, I encourage you to revise your manuscript, accordingly, taking into account all of the concerns raised by the reviewers.

Both reviewers found your manuscript to be well written and well organized. However, each found some methodological limitations, and there is agreement that a few more analyses may help the overall study by showing CASPredict outperforms similar approaches.

Please address all of these issue raised by both reviewers.

I look forward to seeing your revision, and thanks again for submitting your work to PeerJ.

Good luck with your revision,

-joe

Reviewer 1 ·

Basic reporting

CASPredict is an interesting work, as mentioned by the authors, there are relatively few computational methods designed for identifying Cas proteins. This work can be a valuable addition to the research community. In general, the manuscript is well written and easy to follow. However, I see more work and efforts are needed before being accepted for publication.

Experimental design

The research question is well defined. Indeed, we do need such tool for better and accurate prediction of Cas proteins.

Validity of the findings

See the comments in General comments for the author

Additional comments

(1) The result page returned from CASPredict is too simple to be practically useful. Query Sequences, Length, Probability and Yes/No are the only columns returned from the prediction. The author needs to study Macsyfinder (https://journals.plos.org/plosone/article?id=10.1371/journal.pone.0110726) carefully, which provides more detailed information about the Cas protein categories, components, query similarity to core or ancillary components and so on. Accordingly, introduction part needs to provide literature review information about Cas protein categorizations and components. Also, the categorization and component information of the 293 manually reviewed Cas proteins in Uniprot will be interesting to see.
(2) In terms of the accuracy (e.g., 89.68%) of the SVM model, there appears to be more room for improvement. One improvement that authors can do is to use another machine learning approach to do feature selection. For example, random forest is used for feature selection in CRISPR-DT (https://academic.oup.com/bioinformatics/article-abstract/35/16/2783/5273481?redirectedFrom=fulltext) MSIpred (https://www.nature.com/articles/s41598-018-35682-z). Comparison of the current feature selection method with Random Forest method in terms of the model accuracy will be interesting.
(3) In the section of Materials and Methods,
“After excluding the sequences containing ambiguous residues (“X”, “B” or “Z”) or sequence similarity more than 30%, we obtained 155 Cas protein sequences as positive 91 samples”. “Select the non-Cas proteins that sequence similarity less than 40% compared to positive 96 samples”. The 30% vs 40% lack clear justification and citation. Why not other percentage instead.
(4) Authors also need to think about allowing users to upload single bacterial or archaeal genome sequences in FASTA file so that predicted Cas proteins can be provided. If this function can be implemented, it will dramatically enhance the usability of CASPredict.

Reviewer 2 ·

Basic reporting

This manuscript reports a machine learning tool to predict Cas proteins for CRSPR-cas systems. The tool was tested on a training set and an independent set containing both positive and negative cases. The author presented the classification performance.

Experimental design

Questions related to the experimental design:

The method has the following two steps to handle training data:

(iv) select the non-Cas proteins that sequence similarity less than 40% compared to positive samples; (v) keep the number of negative samples and positive samples the same. After the above steps, we finally collected 155 Cas proteins and 155 non-Cas proteins.

Labelled data samples are precious for any learning algorithms, The author did not explain why discard those negative samples, and why select only 155 non-Cas proteins, and why keep the strict balance of the positive and negative class. Similar concerns for the selection of test samples.

The feature selection process also contains a flaw. In 10-fold cross validation, separate feature selection should be applied for each fold of training data. That means the features determined by each fold of training data may be quite different. However, the author just did one-off feature selection for the whole training data set.

Validity of the findings

It’s best to conduct a leave-one-cross-validation process to evaluate the performance.

As the comparison results on the independent test show no difference between the performance of CASPredict and HMMCAS, it is suggested for the author to test the two tools on a new set of proteins.

Additional comments

The manuscript is well organized, but the technical concerns should be technically addressed.

---

## Round 0.2 · Major Revisions

Dear Drs. Yang and colleagues:

Thanks for revising your manuscript based on the concerns raised by the reviewers. Unfortunately, the reviewers still have concerns with your work. It appears that some of their original concerns were not addressed in your revision. This is unfortunate. Also, it appears that a major revision of the tool is needed. The major argument is that in comparison of other tools such as Macsyfinder, this tool does not provide more useful information, even though it adopts a different machine learning technology (SVM). Please consider the limitations of your tool and try to improve it, making a solid argument for its usefulness to the community. Please also address the methodological shortcomings and ambiguity in comparisons noted by reviewer 2.

Please address all of these issue raised by both reviewers.

I look forward to seeing your revision, and thanks again for submitting your work to PeerJ.

Good luck with your revision,

-joe

Reviewer 1 ·

Basic reporting

The authors have addressed some of my comments and suggestions in their revision. Literature references are much better now.

Experimental design

The authors also improve their training procedures (e.g., feature selection and cross-validation strategies) in terms of the comments from both reviewers.

Validity of the findings

They have used new independent testing data to show the performance and accuracy of their tool, which is not so impressive.

Additional comments

The authors have addressed some of my concerns and comments, but they avoid the critical one. Although they modified the manuscript to describe Cas protein categories and components, but their claim “If we understood correctly, the categories, components, and other information mentioned in this article is for the CRISPR-Cas systems but not for the Cas proteins” is totally wrong. CRIPSR-Cas system is composed of two parts: Cas proteins systems and adjacent CRISRP repeats. Essentially, the Macsyfinder paper was to focus on the categories and component information of Cas proteins system, as well as identification of their homologous in a given genome. They adopted Hidden Markov model in training protein sequences profiles and give prediction of the existence possibility of some sort of Cas protein system in a given genome. Now, let compare CASPrdict: it adopts different Machine learning technology - support vector machine and use the optimal dipeptides in training and only answer the question whether a given query protein sequence is a Cas protein or not (possibility).
The authors argued that “The main purpose of developing CASPredict is to facilitate the discovery of potential Cas proteins from unknown protein sequences …”. From biological perspective, any potential CRISPR Cas proteins must have shared core, essential functions while having mutations and other meaningful variations. Unless the authors have the major revisions of their tool, I do not see much impact and usage in the research field. Can authors add new model in tool to answer the questions “Is the given query protein a core protein of Cas system (possibility)”? Can authors compare their result with Macsyfinder result in terms of accuracy, sensitivity or other measures?
Also in the revision, “The core Cas proteins provide a strong barrier for prokaryotic species to resist foreign nucleic acid molecules” is problematic in its interpretation.

Reviewer 2 ·

Basic reporting

The author conducted a revision for the manuscript using the comments and suggestions from the reviewers. Some of my comments have been taken, but the other comments have not been considered so far.

Experimental design

The design for the leave-one-out-cross-validation is not clear. It's suggested for the author to find a new data set for a performance comparison between the proposed method and HMMCAS (method mentioned in the introduction). It seems that the author misunderstood this point.

Validity of the findings

The performance comparison with HMMCAS is necessary as it's an existing Cas protein prediction method with similar prediction performance as the author's approach. The current response to the comments is not satisfactory.

Additional comments

please explain why leave-one-out-cross-validation performance is lower than 10-fold cross validation results. Usually the former is better as more training data is used for prediction.

more comparison with HMMCAS would make the manuscript stronger.

---

## Round 0.3 · Minor Revisions

Dear Drs. Yang and colleagues:

Thanks for revising your manuscript. The reviewer is very satisfied with your revision (as am I). Great! However, there are some minor concerns still raised, and some edits to make. Please address these ASAP so we may move towards acceptance of your work.

-joe

Reviewer 1 ·

Basic reporting

The authors have put good efforts to address my comments and concerns. This version of manuscript is much better than the last version.

Experimental design

Experimental design is good and repeatable.

Validity of the findings

Validation of the result is acceptable

Additional comments

Here are the suggestions or comments for minor revisions:

# 1: Need to define clearly what is the whole dataset, in contrast with training data and testing data.
# 2: In line 112, "select the non-Cas proteins that sequence similarity less
113 than 40%" has grammatical errors.
# 3: In line 149, need to cite the source program for "grid.py"
# 4. In line 210, "the predicted positive samples are ahead of the negative samples", this is unclear in its meaning.

# 5. For method part, these are the 193 dipeptide features that were calculated with the highest F score? What was the cutoff for this? Should this be mentioned under feature selection?


# 6. “The final classification model was based on the whole dataset which could make use of the dataset. The F-score of each feature was calculated based on the whole dataset, and a grid search strategy was also applied on the whole dataset to seek for the best feature number, the error factor c and kernel function variance gamma.”

Not entirely sure what this means. Does it mean that you recalculated the F-score of the dipeptide features on the whole dataset instead of just the training dataset? If so, what did you do with that information? → Was anything changed because of this? If you changed the dipeptide selection, then why did you not just run it with the whole dataset earlier? I think this is fine though because you essentially are running an nested cv loop to find the best parameters.

# 7. I read the discussion where it addressed creating a dataset. Would it not be better to “balance” (at least try to but I can see where manually reviewed is a difficult part as it’s not guaranteed to be a non-cas protein) a whole entire dataset then split it respectively between training and testing sets? If you created the independent dataset the same way as you created your training set, then the training set would have a bias for being positive, hence it also being flawed? Please clarify.

---

## Round 0.4 · accepted · Accept

Dear Drs. Yang and colleagues:

Thanks for revising your manuscript based on the concerns raised by the reviewers. I now believe that your manuscript is suitable for publication. Congratulations! I look forward to seeing this work in print, and I anticipate it being an important resource for groups studying informatics approaches to identifying Cas proteins. Thanks again for choosing PeerJ to publish such important work.

Best,

-joe